# Effect of Pyrolysis Temperature on the Sorption of Cd(II) and Se(IV) by Rice Husk Biochar

**DOI:** 10.3390/plants11233234

**Published:** 2022-11-25

**Authors:** Zheyong Li, Qu Su, Luojing Xiang, Yajun Yuan, Shuxin Tu

**Affiliations:** 1Hubei Provincial Academy of Eco-Environmental Sciences, Wuhan 430072, China; 2State Key Laboratory of Soil Health Diagnosis and Green Remediation for Environmental Protection, Wuhan 430072, China; 3School of Resources and Environmental Engineering, Wuhan University of Technology, Wuhan 430070, China; 4Hubei Urban Construction Design Institute Co., Ltd., Wuhan 430051, China; 5College of Resources and Environment, Huazhong Agricultural University, Wuhan 430070, China; 6Hubei Research Centre for Environment Pollution and Remediation, Wuhan 430070, China

**Keywords:** agricultural waste, biochar, sorption, anions and cations

## Abstract

This study investigated the removal of metal cations (Cd(II)) and metalloid anions (Se(IV)) from their aqueous solution by using agricultural waste (rice husk biochar). Rice husk biochar samples were prepared under 300, 500, and 700 °C pyrolysis conditions and their physicochemical properties were characterized. Aqueous Cd(II) and Se(IV) sorption kinetics and isotherms of rice husk biochar were studied. The results showed that the yield of rice husk biochar decreased from 41.6% to 33.3%, the pH increased from 7.5 to 9.9, and the surface area increased from 64.8 m^2^/g to 330.0 m^2^/g as the pyrolysis temperature increased from 300 °C to 700 °C. Under the experimental conditions, at increasing preparation temperatures of rice husk biochar, the sorption performance of Cd(II) and Se(IV) was enhanced. The sorption capability and sorption rate were considerably higher and faster for Cd(II) ions than for Se(IV) ions. Cd(II) sorption was found to reach equilibrium faster, within 150 min, while Se(IV) sorption was slower and reached equilibrium within 750 min. The maximum sorption capacities of cadmium and selenium by rice husk biochar were 67.7 mg/g and 0.024 mg/g, respectively, according to Langmuir model fitting.

## 1. Introduction

With the rapid expanse of population and development of industrialization, heavy metals (HMs) pollution due to anthropogenic activities has become a serious threat to human health and environmental safety [1]. Cadmium in the aqueous environment usually exists as Cd(II) and is classified as the most ssignificant metal pollutant in China, due to its mobility and toxicity [2]. It has been reported that long-term consumption of drinking water with excessive Cd can lead to respiratory, liver, and kidney toxicity, as well as osteoporosis and bone softening [3]. Selenium, as a metalloid element with similar properties to heavy metals, has significant heterogeneous distribution in the natural environment. For example, in the San Joaquin Valley, California, water selenium concentrations may range from 0.14 to 1.4 mg/kg, resulting in a threat to human health [4]. The World Health Organization (WHO) has estimated a maximum ‘safe’ concentration for Se ions at 40 μg/L in drinking water, while the guideline ratio of Cd(II) is restricted to 1 μg/L [5].

Among current methods for the removal of these two toxic ions from polluted water, sorption is an economical, efficient, and environment-friendly technique. Therefore, finding an appropriate adsorbent with high efficiency and low cost is a current focus of research in this field [6]. According to research, biochar as a sorption material can effectively remove HMs from contaminated water due to its highly porous structure, large specific surface area, and diverse functional groups [7]. For example, Yang et al. (2021) [8] found that biochar derived from Robinia pseudoacacia and durian shells showed high affinity for Cd^2+^, with maximum sorption capacities of 11.4 and 37.6 mg/g, respectively.

In the process of biochar preparation, the pyrolysis conditions, especially pyrolysis temperature, are of great theoretical and practical importance to the physiochemical properties of biochar and its sorption performance for HMs [9]. When biochar is prepared at lower temperatures, the material contains a higher number of active functional groups, which may favor the sorption of pollutants. As the pyrolysis temperature increases, the surface area of the biochar increases, and macropores are destroyed with more small pores forming, which may also favor the sorption of heavy metals. In a recent study, increasing the pyrolysis temperature led to an increase in the pH, aromaticity, and surface area of the biochar [10], improving the removal capacity of biochar for Cd^2+^ [5]. Lehmann [11] indicated that the surface area of woody biochar increases with increasing production temperature from 300 to 500 °C, but decreases from 500 to 800 °C. Therefore, it is important to understand the relationship between pyrolysis temperatures and the sorption properties of biochar as a sorption material for HMs.

As a type of biochar, rice husk biochar is derived from the combustion and pyrolysis of widely produced rice by-products. Rice is planted globally and is a staple food for nearly 3 billion people [12]. Due to the considerable calorific value of rice husk (16.7 × 10^6^ J/kg), a large amount of rice husk is used annually as biomass fuel for power generation [13]. The resulting rice husk biochar is a carbon-rich and silica-rich porous solid residue. It is similar to activated carbon, with large specific surface area, high porosity, porous structure, and abundant functional groups, and is a highly promising adsorbent material [14]. Due to its negative surface charge, it usually has a good sorption effect on cations in water. However, there have been few studies on rice husk biochar for sorption of anions in water, especially comparing differences in anion and cation sorption [13,15].

It is important to explore other potential uses of rice husk biochar, including the production of biochar at different temperatures as a low-cost adsorbent for removal of pollutants. Accordingly, the objective of this work was to investigate the changes in the microstructure and physicochemical properties of rice husk biochar (RHB) obtained at different pyrolysis temperatures, and the sorption properties and mechanisms of RHB for cation (Cd(II)) and anion (Se(IV)). RHB samples were prepared from rice husk at pyrolysis temperatures of 300, 500, and 700 °C. In addition, the sorption kinetics and isotherms of RHB for Cd(II) and Se(IV) were also investigated. The results provide a reference for the preparation of RHB with different properties and for the detoxification of Cd(II) and Se(IV) in wastewater.

## 2. Materials and methods

### 2.1. Materials

Rice husk was obtained from Jingmen Thermal Power Plant in Hubei, China. It was sieved, rinsed, and then macerated with 1.0 mol/L HCl solution in a solid–liquid ratio of 1:10 for 2 h. The rice husk was washed three times with deionized water and dried at 60 °C for 24 h. The dried sample was crushed and passed through a 20-mesh sieve.

The rice husk biochar (RHB) was prepared in a high-temperature chamber furnace (BR 12N, Bona Thermo, Shanghai, China) with a volume of 16 L. The charcoal was extracted from 10 g of RHB after pretreatment. The pyrolysis temperature was set at 300, 500, or 700 °C for 10 g of pretreated rice husk, and the temperature was increased by 5 °C/min for 2 h. The resulting RHB was ground and sieved through a 60-mesh sieve and labeled as R300, R500, or R700, respectively.

### 2.2. Experiment Design

#### 2.2.1. Sorption Kinetics

The RHB prepared under different conditions was weighed out at 0.2 g (2 g/L) into a 2500 mL conical flask with 100 mL of CdCl_2_ solution (100 mg/L, pH = 7) or Na_2_SeO_3_ solution (1 mg/L, pH = 5). Each treatment was repeated three times, and the suspensions were taken after shaking times of 1/6, 1/3, 0.5, 1, 3, 5, 12, 24, and 27 h on a shaker at 25 °C. The suspensions were filtered through a syringe (filter pore size = 0.22 μm). The solutions were stored at 4 °C and the content of Cd(II) and Se(IV) in the filtrate was determined. The removal rate during the process was calculated as:Removal% = (C_i_ − C_t_)/C_i_ * 100%(1)
where C_i_ and C_t_ represent the initial concentration of the solution and the concentration at time t (mg/L), respectively.

#### 2.2.2. Equilibrium Experiments

Equilibrium experiments were carried out in a 100 mL centrifuge tube: the solid–liquid ratio of the reaction system was 2 g/L. For example, 0.1 g of rice husk charcoal was mixed with 50 mL of CdCl_2_ solution (pH = 7) or Na_2_SeO_3_ solution (pH = 5) containing different concentrations of Cd(II) (10, 50, 100, 200, 500 mg/L) or Se(IV) (0.01, 0.05, 0.1, 0.2, 0.5 mg/L). The suspension was shaken for 6 h at 25 °C and then centrifuged at 4000 r/min for 10 min to separate the supernatant and the solid residue. The supernatant was filtered and stored at 4 °C for measurement.

The concentration of Cd(II) in the solution was determined by atomic absorption spectrometry (AAS 240-Agilent Technologies, Beijing, China), and the concentration of Se(IV) was determined by hydride generation atomic fluorescence spectrometry (AFS-8220, Jitian, Beijing, China).

### 2.3. Characterizations

The pH of rice husk biochar was determined by adding 1.0 g biochar into the centrifuge tube (50 mL), and equilibrating in suspension at a ratio of 1:20 (*w*/*v*) for 1 h; the pH value of the suspension was detected by pH meter (PHS-3C, Shanghai, China) to give the pH of the RHB. Cation exchange capacity (CEC) were measured by a modified NH4-acetate compulsory displacement method [16]. The C, H, and N contents of the samples were tested with an elemental analyzer (Vario PYRO cube and Isoprime 100, Elementar, Germany), while O was obtained by subtracting the C, H, N, and ash content. Phase composition analysis of all samples was measured by a Bruker D8, Germany X-ray diffractometer (XRD) with CuKα (λ = 0.15418 nm) radiation at 40 kV, 40 mA. The specific surface area (SSA) was estimated by an automatic specific surface analyzer (ASAP2460, Micromeritics, Norcross, U.S.A.). Rice husk biochar was weighed (0.1~0.2 g) and degassed at 110 °C for 3 h. The specific surface area was calculated according to the multi-point Brunauer–Emmett–Teller (BET) method. Scanning electron microscopy (SEM) (VEGA 3 XMU, Tescan, Shanghai, China) was employed to observe the morphological structure of the rice husk biochar. The surface morphology of the samples was tested after removing a small amount of rice husk biochar on the surface of a conductive tape. The accelerating voltage of the test was 5 kV and the current was 5 μA. The IR absorption spectra of the rice husk biochar were analyzed on a Fourier transform infrared spectrometer (Equinox 55, Bruker, Karlsruhe, Germany) at room temperature. An appropriate amount of dried sample was mixed well in an agate mortar with anhydrous KBr at a weight ratio of 1:100. The wave number range for the test was from 4000 to 400 cm^−1^ with a scan number of 64 and a resolution of 4 cm^−1^.

### 2.4. Analytical Methods

#### 2.4.1. Sorption Isotherm Models

(1) Langmuir model
(2)$$q=q_{m}*K_{L}*C/\left(1+K_{L}*C\right)$$
where q (mg/g) represents the equilibrium sorption capacity, qm (mg/g) represents the calculated maximum sorption capacity. K_L_ (L/mg) is the sorption constant of the Langmuir model and C (mg/L) represents the concentration of adsorbate at equilibrium.

(2) Freundlich model
(3)$$q=K_{f}C^{1/n}$$
where q (mg/g) and C (mg/L) have the same meaning as in Equation (1). K_f_ and n are empirical constants, where K_f_ is the sorption performance index and n is the dimensionless exponent of the reaction bond energy.

#### 2.4.2. Sorption Kinetic Models

(1) Pseudo-first-order model
(4)$$q_{t}=q_{e}\left(1-e^{-k_{1}t}\right)$$
where q_t_ and q_e_ represents the sorption capacity (mg/g) at time t and equilibrium, respectively, k_1_ (h^−1^) is the rate constant of the pseudo-first-order model.

(2) Pseudo-second-order model
(5)$$t/q_{t}=1/{k*q}_{e}^{2}+t/q_{e}$$
where q_t_ and q_e_ have the same meaning as in Equation (3), k_2_ (h^−1^) is the rate constant of the pseudo-second-order model.

#### 2.4.3. Data Analysis

The experimental data were compiled using Excel 2018, and Origin 2018 was used for model fitting and graphing.

## 3. Results

### 3.1. Characteristics of RHB

The yield and basic properties of rice husk biochar in this experiment are summarized in Table 1. It was observed that as the production temperature increased from 300 to 700 °C, the yield, elemental content (C, H, O) and cation exchange capacity (CEC) of rice husk biochar decreased, while the pH, Si content, and surface area increased. The rise in pH of the rice husk biochar slowed down after 500 °C, while the surface area continued to increase. The surface area of R700 reached 5.16 times that of R300. The H/C value indicates the aromaticity of the material, and the O/C value represents the abundance of functional groups. As the preparation temperature increased, the number of functional groups of RHB decreased and the aromaticity decreased.

The XRD patterns and FTIR spectra of rice husk biochar prepared at different temperatures showed that changing the pyrolysis temperature had little effect on the mineral composition of RHB, but the number and type of surface functional groups changed (Figure 1). In the XRD patterns, no other distinctive characteristic peaks were observed, indicating that they were amorphous carbon or silica structures (Figure 1a). In the FTIR spectra (Figure 1b), all RHBs had a C=C peak at 1067 cm^−1^, which is typical for biochar and represents its aromatic structure. The peak at 1701 cm^−1^ observed in R300 and R500 could be ascribed to C=O vibration, and represents mainly carboxyl groups. Such a peak was not observed in R700 because the increasing pyrolysis temperature led to the decomposition of the carboxyl groups in the biochar. Peaks of 1070 and 796 cm^−1^ were present in all RHBs and represent Si-O-Si and Si-O, respectively, which may be due to the abundance of Si in RHB.

### 3.2. Sorption Efficiency

#### 3.2.1. Cd Sorption

Batch experiments were conducted to investigate the sorption kinetic characteristics of Cd(II) sorption by rice husk biochar prepared at different temperatures (Figure 2). The sorption rate of Cd(II) by RHB was relatively fast during the first 150 min, reaching 51.2%~71.1% of the maximum sorption amount in a short time. Subsequently, the sorption rate slowed during the 150 min~300 min phase, reaching 64.5~89.2% of the maximum sorption, after which sorption equilibrium was slowly reached. The simulation results of the sorption kinetic curves are shown in Table 2, revealing that the pseudo-second-order kinetic model with R^2^ of 0.906~0.973 could describe the data better than the pseudo-first-order kinetic model, which indicated that the sorption process of Cd(II) by RHB was mainly based on chemical sorption.

The isothermal sorption data for Cd(II) (Figure 2) showed that the sorption capacity of rice husk biochar increased with the rise of initial Cd(II) concentration. The sorption capacity followed the following trend: R700 > R500 > R300. The results of the Langmuir model had a better fit than the Freundlich model for the three sets of sorption data, with R^2^ > 0.935 (Table 2), indicating a single molecular layer sorption process of Cd(II) in solution by RHB. From the fitting parameters, it was revealed that the maximum sorption amounts (q_m_) of Cd(II) by R300, R500, and R700 were 4.15, 58.5, and 67.7 mg/g, respectively.

#### 3.2.2. Se(IV) Sorption

The sorption kinetic data of Se(IV) shown in Figure 3a were subjected to pseudo-first-order (PFO) and pseudo-second-order (PSO) fitting, to investigate the sorption process. The regression parameters for Se(IV) sorption kinetics are shown in Table 3. The sorption of Se(IV) reached equilibrium at 750 min. The PSO model was more suitable than the PFO model for fitting the sorption process, with R2 ranging from 0.831 to 0.973, indicating the combined effect of multiple sorption processes. The maximum equilibrium sorption of R700 (0.128 mg/g) was much higher than that of R500 or R300 (0.0304 and 0.0132 mg/g). However, the sorption of Se(IV) by R700 was the last to reach equilibrium, and the sorption rates (k_1_ and k_2_) of R700 were the lowest among the three RHBs.

To provide insight into the potential sorption mechanisms involved, the sorption isotherms of Se(IV) by RHB produced at different preparation temperatures were investigated (Figure 3b). The sorption capacity of RHB for Se(IV) gradually increased with the rising preparation temperature, i.e., R700 > R500 > R300. Langmuir and Freundlich models were used to fit the isothermal sorption data. The results showed that Langmuir model could better describe the sorption process of Se(IV), with R^2^ of 0.649~0.951 (Table 3), which indicated a monolayer sorption of Se(IV) by RHB. The Langmuir maximum sorption capacities (Q_max_) of R300, R500, and R700 on Se(IV) were 0.0031, 0.011, and 0.0241 mg/g, respectively.

## 4. Discussion

### 4.1. Effect of Pyrolysis Temperature on the Properties of RHB

In this experiment, the RHB yield decreased as the preparation temperature increased (Table 1). This may be attributed to the fact that increasing the production temperature promotes the formation of bio-oil and synthesis gas, leading to a reduction in solid product yield [17]. It is worth mentioning that the yield of rice husk biochar is higher than that of other sources of biochar, due to the high content of silica in rice husk [18]. The pH of RHB increases with the formation of alkaline substances and the volatilization of acidic substances [19]. Hemicellulose, cellulose, lignin, and other components of the rice husk undergo dehydration, dihydroxylation, and other chemical reactions to produce acidic substances [20]. The pH of R300 (7.5) was almost neutral because there remined a certain amount of acidic functional groups in the biochar at this temperature [19]. R500 and R700 were significantly alkaline, which was due to the increased volatilization of highly acidic substances at high temperatures. We also observed that the acidic and alkaline contents of RHB changed significantly in the pyrolysis temperature range of 300–500 °C. Continuing to increase the production temperature only slightly increased the pH of rice husk biochar [18]. Therefore, pyrolysis temperature had a significant effect on the yield and pH of biochar.

Increasing the pyrolysis temperature accelerates the pyrolysis and volatilization of rice husk, leading to an increase in surface area, which contributes to an increase in sorption capacity. When heated to 300 °C, the surface area of RHB (R300) was low (64.78 m^2^/g). Upon further heating, porous structures were created in the biochar due to volatilization of organic residues, resulting in a significant increase in the surface area of RHB (R700) (330.0 m^2^/g) [21]. The C content of all three RHBs ranged from 51.60–52.78%. As the production temperature increased from 300 °C to 700 °C, the C content of the biochar increased slightly. At the same time, the content of H, N, and O in the material decreased due to the dehydration of volatile organic matter [22]. Correspondingly, the O/C and H/C values of RHB decreased at higher pyrolysis temperatures, indicating that the hydrophilicity of the biochar surface decreased, the aromaticity increased, and its structure became more stable [23].

### 4.2. Effect of Pyrolysis Temperature on the Sorption Performance

The trend of Cd(II) sorption by each of the three RHBs was consistent regardless of the pyrolysis temperature. It was a rapid sorption at first, followed by a slowing of the sorption rate, and reached equilibrium after 6 h (Figure 2a). This was due to the driving force for mass transfer of the Cd(II) between the liquid and solid phases (i.e., Cd(II) solution and rice husk biochar), and the sorption sites available on the surface of the biochar [24]. At the beginning of the sorption, the Cd(II) concentration gradient between the two phases was large, resulting in a larger driving force for mass transfer and a high rate of sorption. As sorption proceeded, the Cd(II) concentration gradient decreased and the sorption sites on the surface of rice husk biochar became saturated. At this point, the effects of film diffusion and intra-particle diffusion were becoming dominant, the sorption rate slowed, and there was a gradual convergence to sorption equilibrium [25]. The kinetic fitting results showed that PSO kinetics better modeled the sorption process for R700 and R500. In general, the kinetics of sorption reactions dominated by chemical processes are usually very fast [26,27]. In this experiment, the sorption rate (k_2_) of rice husk biochar decreased and the equilibrium time increased with the rise of pyrolysis temperature. This indicates that the removal of Cd(II) by rice husk biochar changed from a predominantly chemical process to a combination of physical and chemical processes [28]. This may be attributed to the decrease in the number of functional groups of rice husk biochar due to increasing the pyrolysis temperature (Table 1), leading to weaker chemisorption. Importantly, the Langmuir model revealed that the maximum sorption qm of R700 was greater than that of R500 or R300 (Table 2), which could be attributed to the larger surface area and higher pH of the rice husk biochar produced under high-temperature pyrolysis. The possible mechanisms include precipitation, electrostatic ion exchange, and complexation of oxygen-containing functional groups with Cd(II) [29]. The aromatic structure of rice husk biochar can act as an electron donor for cation-π interaction with Cd(II) [30]. In comparison with other studies, it was found that the maximum sorption (67.7 mg/g) observed in the present study was higher than most other biochars of plant origin (e.g., bagasse, peanut shells, pine bark, oak bark, wolfberry bark, or rapeseed stalk) [31,32,33]. Hoyos-Sánchez et al. [34] observed a maximum sorption of 28.27 mg/g of Cd(II) for alkali-treated rice husk biochar, but did not mention the production temperature. Huang et al. [35] found a maximum sorption of 56.34 mg/g of Cd(II) for rice husk biochar produced at 700 °C. A series of factors such as different rice growing regions and production conditions may lead to differences in the maximum sorption capacity of rice husk biochar. In conclusion, rice husk biochar has a strong sorption effect on Cd and is worthy of further study.

Compared with Cd(II), the kinetics of Se(IV) sorption by rice husk char are relatively slow and the capacity for sorption is much smaller (Table 2). The kinetic fitting results showed that both PFO and PSO kinetics were able to fit the results well. Compared with Cd(II), more physical processes (film diffusion and intra-particle diffusion) are involved in the sorption of Se(IV) by rice husk biochar. The sorption rate decreased with increasing pyrolysis temperature, which was similar to the sorption of Cd(II). The Langmuir model was able to describe the experimental data better than the Freundlich model, indicating that the sorption of Se(IV) by rice husk biochar was mainly monolayer sorption [36]. Increasing the pyrolysis temperature increased the maximum sorption of Se(IV) (Table 3), which was also similar to the results for Cd(II). Previous studies have shown that unmodified biochar has a weak sorption capacity for anions such as As(V) and Se(IV) in solution [15]. Clemente et al. [15] investigated the sorption of Se(IV) by various types of biochar. The results indicated that softwood biochar had the highest sorption capacity (0.2 mg/g), while hardwood (0.02 mg/g), hay (0.07 mg/g), and chicken manure biochar (0.05 mg/g) each had lower sorption capacity, comparable to the results of this study (0.02 mg/g). Therefore, the sorption of Se(IV) by rice husk biochar was not effective. It is necessary to improve the sorption capacity by appropriate modification methods [36].

## 5. Conclusions

This study showed that the pyrolysis temperature influenced the properties of rice husk biochar and thereby differently affected the Cd(II) and Se(IV) sorption processes. Under the presented experimental conditions, the yield of rice husk biochar gradually decreased with increasing preparation temperature. The pH, surface area, and aromaticity of RHBs increased, their structures became more stable, and their polarity decreased. The sorption capacity for Cd(II) and Se(IV) by rice husk biochar increased with the increase of pyrolysis temperature. It is obvious that the large specific surface area of RHB played an important role in the sorption. Pseudo-second-order kinetics and the Langmuir model were well able to describe the data for sorption of Cd(II) and Se(IV) by rice husk biochar. The analysis revealed that the surface of rice husk biochar was rich in functional groups. With the increase of pyrolysis temperature, the absorption bands of some functional groups changed, which led to the increase of surface area and finally affected the sorption capacity for Cd(II) and Se(IV). However, the removal rate and sorption capacity for Se(IV) anions were significantly lower.

## Figures and Tables

**Figure 1 plants-11-03234-f001:**
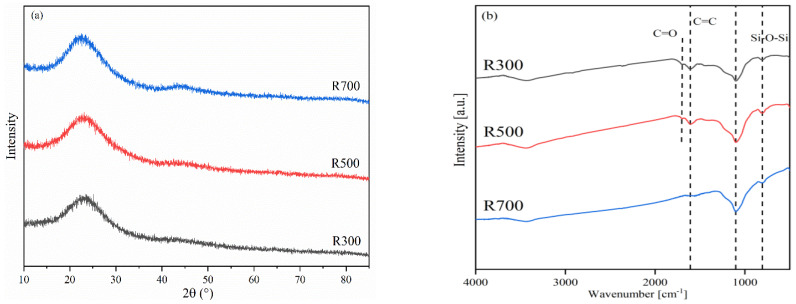
(**a**) The XRD patterns and (**b**) FTIR spectra of RHB produced at different temperatures.

**Figure 2 plants-11-03234-f002:**
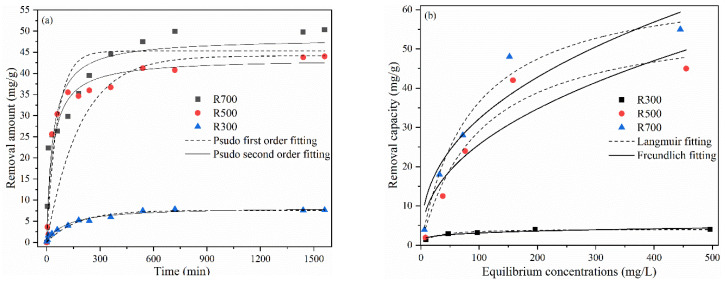
(**a**) Kinetic analysis for Cd^2+^ sorption by RHB (20 mL 100 mg/L Cd(II) mixed with 0.2 g RHB, pH = 7); (**b**) Sorption isotherm of RHB on Cd(II) (reaction conditions: 20 mL 10~500 mg/L Cd(II) solution mixed with 0.2 g RHB, pH = 7).

**Figure 3 plants-11-03234-f003:**
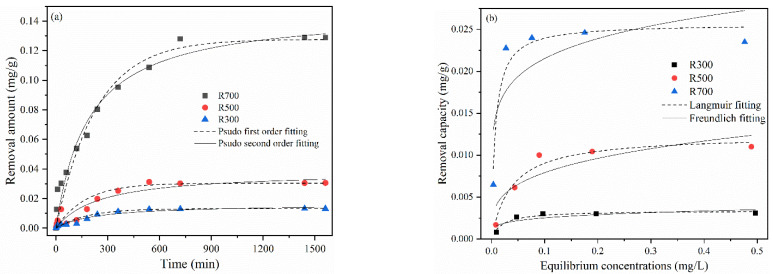
(**a**) Kinetics results for the removal of Se(IV) (20 mL 1 mM Se(IV) mixed with 0.2 g biochar, pH = 5); (**b**) Isotherms results for the removal of Se(IV) (20 mL Se(IV) solution mixed with 0.2 RHB).

**Table 1 plants-11-03234-t001:** The yield, pH, surface area, and elemental composition of rice husk biochar produced at different temperatures.

Properties		R300	R500	R700
Yield (%)		41.60	37.90	33.30
pH		7.5	9.3	9.9
BET surface area (m^2^/g)		64.78	257.5	330.0
Elemental composition (%)	C	51.6	52.78	52.54
	H	2.76	2.07	1.012
	N	0.09	0.07	0.11
	O	16.5	11.2	8.98
H/C		0.053	0.039	0.019
O/C		0.321	0.212	0.170
Si (g/kg)		145.6	165.2	195.5
CEC (cmol/kg)		45.62	43.15	39.08

**Table 2 plants-11-03234-t002:** Parameters of kinetics and isotherms for the removal of Cd(II) by RHB produced at different temperatures.

**Samples**	**Pseudo-First Order**			**Pseudo-Second Order**		
	**q_e_ (mg/g)**	**k_1_ (h^−1^)**	**R_2_**	**q_e_ (mg/g)**	**k_2_ (h^−1^)**	**R^2^**
R700	45.28	0.990	0.804	48.27	0.0363	0.906
R500	44.15	0.316	0.649	43.45	0.0401	0.959
R300	7.57	0.348	0.951	8.360	0.0648	0.973
**Samples**	**Langmuir**			**Freunlich**		
	**q_m_ (mg/g)**	**K_L_ (L/mg)**	**R^2^**	**K_f_ (mg/g)**	**1/n**	**R^2^**
R700	67.7	0.012	0.974	4.87	0.409	0.881
R500	58.5	0.010	0.935	3.43	0.436	0.807
R300	4.15	0.052	0.961	1.18	0.211	0.831

**Table 3 plants-11-03234-t003:** Parameters of kinetics and isotherms for the removal of Se(IV) by RHB.

**Samples**	**Pseudo-First Order**			**Pseudo-Second Order**		
	**q_e_ (mg/g)**	**k_1_ (h^−1^)**	**R^2^**	**q_e_ (mg/g)**	**k_2_ (h^−1^)**	**R^2^**
R700	0.128	0.258	0.957	0.147	2.16	0.969
R500	0.0304	0.3516	0.831	0.038	6.12	0.876
R300	0.0132	0.3354	0.932	0.016	15.66	0.948
**Samples**	**Langmuir**			**Freunlich**		
	**q_m_ (mg/g)**	**K_L_ (L/mg)**	**R^2^**	**K_f_ (mg/g)**	**1/n**	**R^2^**
R700	0.0241	0.139	0.951	0.0305	0.151	0.473
R500	0.011	0.026	0.649	0.0151	0.281	0.717
R300	0.0031	0.051	0.804	0.00403	0.203	0.576

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
