# Peer review of "Effect of Pyrolysis Temperature on the Sorption of Cd(II) and Se(IV) by Rice Husk Biochar"

_plants, 2022, doi:10.3390/plants11233234_

Round 1

Reviewer 1 Report

Nice experimental work, with classic statistical modeling.

Author Response

Nice experimental work, with classic statistical modeling.
"I still keep my current Comments and Suggestions for Authors: "Nice experimental work, with classic statistical modeling", here "nice" meaning "conventional", because the paper reports an ordinary experimental work having two parts: the first, in which rice husk biochar was prepared under 300, 500 and 700 ℃ pyrolysis (followed by the routine physicochemical properties characterization) and, then, its adsorption performance of Cd(II) and Se(IV) experimentally tested. Nothing extraordinary here, all the routine steps followed "ad litteram", to be sure the paper gets published. Since no adsorption report gets published without thermodynamic and kinetic part, they regressed two of the the most used thermodynamic (Langmuir and Freundlich) and kinetic (Pseudo-first-order and Pseudo-second-order) models to get their parameters. From their experimental data, it was clear that the adsorption is a saturation type process, for which Freundlich's model doesn't work by definition, but I respected their hard work - sometimes, when the adsorption process is in its infancy, Freundlich's model works fine. Also, they could explain the slower kinetic of Se(IV) vs Cd(II) using the equivalent diameter and its relationship with diffusion, but they are plain experimentalists, doing routine work, checking all the items which could ensure the publication. And, since the technical English was good enough, I considered that it is a "Nice experimental work, with classic statistical modeling". From my point of view, to improve this report, the authors should have to do a simultaneous extraction of  Se(IV) AND Cd(II), and to test the effect of their ratio, but, then, they have to study, as well, the effects of co-ions - but this is a completely different research, while my opinion was formed on what they did."

Reply: Thank you for your serious comment. We do agree with you that a simultaneous adsorption experiment should be done. Actually we are working on this and also developing some modified biochar to perform the test. Hopefully we can have more conversations in the future for this work.

Reviewer 2 Report

Dear authors,

first of all I do not agree thta you measure an adsorption; you measure an absorbtion! Adsorption is an atomic binding! However, be consequent in your writing.

We have not measured and OH groups in oure biochars. But, I think that you have produced some functional groups to the Si concentration in the rice husks! For this you should measure the Si concentration in biochars produced !

What is about the cation and anion exchange capacity of the biochars? Please show the data!

Author Response

Dear authors,

first of all I do not agree thta you measure an adsorption; you measure an absorbtion! Adsorption is an atomic binding! However, be consequent in your writing.

We have not measured and OH groups in oure biochars. But, I think that you have produced some functional groups to the Si concentration in the rice husks! For this you should measure the Si concentration in biochars produced !

What is about the cation and anion exchange capacity of the biochars? Please show the data!

Reply:Thank you for your comment. We performed complementary experiments to determine Si content and CEC of rice husk biochar. Results showed in L156-157 and Table 1 in L165.
Please find the manuscript in the attachment.

Reviewer 3 Report

The manuscript “Effect of pyrolysis temperature on the adsorption characteristics of Cd(II) and Se(IV) by rice husk biochar” investigated  the  removal  of  metal  cations  (Cd(II))  and  metalloid  anions  (Se(IV)) from their aqueous solution by using an agricultural waste (rice husk biochar).

The results showed that the yield of rice husk biochar decreased, while the pH surface area increased. While the subject is both interesting and worth publishing in “plants”, In my opinion, this is a very interesting study, the manuscript is well-structured, but the writing needs to be improved. Here are some issues should to be addressed to be acceptable, please revised the manuscript carefully:

Comments

·         The key words should be distinct from the title words. Authors can change them to be more specific. Please consider updating the keyword list and using synonyms.

·         In the introduction, the authors should give more details regarding rice, please cite the following references:

https://doi.org/10.1071/FP21268

https://doi.org/10.1016/j.plaphy.2021.07.013

·         Please revise the objective and hypothesis of the study at the end of the introduction to be more attractive

·         The authors did not give any details regarding the experimental design, which design used ? How many factors used ? How many levels of factors used in this study  ? .

·         Authors should add one sentence at the end of each paragraph to conclude the whole paragraph to make it easy for the reader

·         The conclusions should be revised to be concise and attractive

Author Response

The manuscript “Effect of pyrolysis temperature on the adsorption characteristics of Cd(II) and Se(IV) by rice husk biochar” investigated  the  removal  of  metal  cations  (Cd(II))  and  metalloid  anions  (Se(IV)) from their aqueous solution by using an agricultural waste (rice husk biochar).

The results showed that the yield of rice husk biochar decreased, while the pH surface area increased. While the subject is both interesting and worth publishing in “plants”, In my opinion, this is a very interesting study, the manuscript is well-structured, but the writing needs to be improved. Here are some issues should to be addressed to be acceptable, please revised the manuscript carefully:

Comments

·         The key words should be distinct from the title words. Authors can change them to be more specific. Please consider updating the keyword list and using synonyms.

Reply: Thank you for your comment. Selenium and cadmium have changed into anions and cations. Rice husk have changed into agricultural waste. L26

·         In the introduction, the authors should give more details regarding rice, please cite the following references:

https://doi.org/10.1071/FP21268

https://doi.org/10.1016/j.plaphy.2021.07.013

Reply: Thank you for your comment. Detail description of the rice has been added to L64-65.

·         Please revise the objective and hypothesis of the study at the end of the introduction to be more attractive

Reply: Thank you for your comment. We tried to add sentences in L74-75 to make it more attractive.

·         The authors did not give any details regarding the experimental design, which design used ? How many factors used ? How many levels of factors used in this study  ? .

Reply: Thank you for your comment. Detailed factors and levels were added in L101 and L111.

·         Authors should add one sentence at the end of each paragraph to conclude the whole paragraph to make it easy for the reader

Reply: Thank you for your comment. A conclusion sentence was added in L248-249, L296-297.

·         The conclusions should be revised to be concise and attractive

Reply: Thank you for your comment. The conclusion was revised in L315.

Please find the manuscript in the attachment.

Round 2

Reviewer 2 Report

Dear collegs,

Thsank you very much for revising the manuscript! However, you still report over an adsorption. However, I think that you measure an absorbtion. Ok, you have measured the CEC! Is this the sum of the cations (Ca, Mg, K, Na) or do you present the Ba exchange in cmol/kg? How have you measured the CEC?

Regards

Author Response

Thank you for your comment. We have thought carefully about this issue and changed all the "adsorption" word into "sorption" in the whole paper to make it more accurate. For CEC determination, we present the sum of Ca, Mg and K. Cation exchange capacity (CEC) were measured by modified NH4-acetate compulsory displacement method (Gaskin et al., 2008).  It was tested by flame specphotometry after NH4-acetate (1mol/L NH4OAc, pH=7.0)  leaching. The revision is shown in L121-122.

Reviewer 3 Report

Thanks for your contribution 

Author Response

Thank you for your time!